# Stereocomplexation Reinforced High Strength Poly(L-lactide)/Nanohydroxyapatite Composites for Potential Bone Repair Applications

**DOI:** 10.3390/polym14030645

**Published:** 2022-02-08

**Authors:** Naishun Guo, Mengen Zhao, Sijing Li, Jiahui Hao, Zhaoying Wu, Chao Zhang

**Affiliations:** School of Biomedical Engineering, Shenzhen Campus of Sun Yat-sen University, Shenzhen 518107, China; guonsh@mail2.sysu.edu.cn (N.G.); zhaomen@mail2.sysu.edu.cn (M.Z.); lisj8@mail2.sysu.edu.cn (S.L.); haojh@mail2.sysu.edu.cn (J.H.)

**Keywords:** polylactide, hydroxyapatite, nanocomposite, stereocomplex, mechanical property

## Abstract

Composite materials composed of polylactide (PLA) and nano-hydroxyapatite (n-HA) have been recognized as excellent candidate material in bone repai The difference in hydrophilicity/hydrophobicity and poor interfacial compatibility between n-HA filler and PLA matrix leads to non-uniform dispersion of n-HA in PLA matrix and consequent poor reinforcement effect. In this study, an HA/PLA nanocomposite was designed based on the surface modification of n-HA with poly(D-lactide) (PDLA), which not only can improve the dispersion of n-HA in the poly(L-lactide) (PLLA) matrix but also could form a stereocomplex crystal with the matrix PLLA at the interface and ultimately lead to greatly enhanced mechanical performance The n-HA/PLA composites were characterized by means of scanning electron microscopy, Fourier transform infrared spectroscopy, X-Ray diffraction, thermal gravity analysis, differential scanning calorimetry, and a mechanical test; in vitro cytotoxicity of the composite material as well as its efficacy in inducing osteogenic differentiation of rat bone marrow stromal cells (rMSCs) were also evaluated. Compared with those of neat PLLA, the tensile strength, Young’s modulus, interfacial shear strength, elongation at break and crystallinity of the composites increased by 34%, 53%, 26%, 70%, and 17%, respectively. The adhesion and proliferation as well as the osteogenic differentiation of rMSCs on HA/PLA composites were clearly evidenced. Therefore, the HA/PLA composites have great potential for bone repai.

## 1. Introduction

As one of the most popular biodegradable polymers, poly(lactic acid)/polylactide (PLA) has attracted tremendous attention as a biomaterial, owing to its excellent biocompatibility and biodegradability [1,2]; efforts have been devoted to its applications as a drug delivery matrix, tissue engineering scaffold, and, in addition, orthopedic implants [3,4]. In order to meet the clinical requirements of long-term orthopaedic implantation, the problems of inadequate mechanical properties and poor osseointegration of PLA need to be solved. Firstly, the brittleness of PLA may increase the risk of mechanical failure [5]. Second, the biological inertness of PLA usually results in low affinity to osteoblast [6]. In addition, the acidic degradation products after implantation could eventually lead to aseptic inflammation at the implantation site [7].

Bioactive inorganic fillers can be used to effectively improve the mechanical strength and the biological activity of PLA [8,9]. Commonly used inorganic fillers include hydroxyapatite (HA), β-tricalcium phosphate (β-TCP), bioglass (BG), and calcium carbonate, etc. [10,11,12]. The bioactivity of PLA composites with these inorganic fillers comes from the dissolution of calcium ion, phosphate ion, and, in addition, silicate ions; these ions can not only act as the calcium source in the mineralization but also promote the proliferation and osteogenic differentiation of osteoprogenitor cells or bone marrow stromal cells (rMSCs) through different signaling pathways [13,14,15,16,17,18]. Furthermore, the nano-sized inorganic fillers are conducive to the interaction between osteoblasts and biomaterial surface The addition of inorganic fillers into the PLA matrix may also produce a rough surface, which favors the adhesion and spreading of the osteoblast [19,20,21]. In addition, the bioactive inorganic fillers, such as HA [22] and BG [23], could induce the onset of crystallization and re-mineralization.

In the case of the composite materials of n-HA and PLA, due to the large difference in hydrophilicity/hydrophobicity between n-HA particles and PLA, prestine n-HA can hardly be dispersed uniformly in the PLA matrix, which makes it difficult to achieve nano-reinforcement [24,25]. In fact, the agglomeration of n-HA in the PLA matrix may pose a detrimental effect on the mechanical properties of the PLA/HA nanocomposite [26]. Classical strategies applied to solve the above problems usually involve the surface modification of n-HA using a silane coupling agent or hydrophobic/non-polar alkyl/polymeric chain Rocha et al. modified the surface of the HA with fatty acids (adipic, sebacic, lauric, and linoleic), which was then compounded with PLLA. It was found that the graft chain length and the presence of double COOH groups were more favorable to improve the interfacial interactions, and the tensile strength increased from 15 to 25 MPa [27]. In addition, supramolecular interaction, hydrogen bonding, and stereocomplexation are also used to enhance the interaction between filler and PLA matrix, strengthen the interface integration, and improve the reinforcement effect [28,29].

The intermolecular interaction between PLA chains has been utilized to improve the interfacial interaction between the n-HA filler and PLA matrix; generally, PLA oligomers are grafted onto n-HA to achieve better dispersion in the PLA matrix and improve the mechanical propertie It has been reported that stereocomplex could form between enantiomeric poly(L-lactide) (PLLA) and poly(D-lactide) (PDLA) in solution, oriented drawing, interface, hot drawing, and even during 3D printing [30,31,32,33,34]; the stereocomplex produces numerous dense, micron-sized crystals that act as cross-linking points between polymer chains and connect the high-density “tie chains” between crystals, thereby significantly improving the mechanical properties [35]. The tensile strength and Young’s modulus of PDLA/PLLA blend films are at least 30% higher than those of the corresponding pure PLLA films in almost any molecular weight rang The presence of stereocomplex crystals in the PLA matrix not only improves the mechanical properties but also enhances the crystallinity and thermal stability [36].

Grafting PDLA chains onto inorganic fillers may enhance the interfacial adhesion between the PLA matrix and inorganic phase [37]. PDLA-grafted magnesium oxide (MgO), titanium dioxide (TiO_2_), ferroferric oxide (Fe_3_O_4_), and clay have been explored to prepare composite materials with PLLA [38,39,40,41]. Kum et al., grafted PDLA oligomers onto the surface of MgO nanorods and then compounded them with PLLA. The tensile strength and modulus of the nanocomposite films increased by 20% due to the formation of the stereocomplex [38]. Huang et al. grafted PDLA and PLLA with different molar masses onto the HA surface and then compounded with the PLLA. The tensile strength increased from 62 to 72 MPa. The strengthening effect was stronger in the case of the PDLA-grafted HA instead of the PLLA with similar molar mas However, the low activity of the hydroxyl groups on the HA surface leads to a low grafting rate (6.2%), which hinders the improvement of mechanical properties [42]. Therefore, improving the grafting rate of the PDLA on the HA surface is a promising strategy to obtain high strength PLLA/HA nanocomposite

In this study, an n-HA/PLA nanocomposite was designed based on the surface modification of n-HA with PDLA and its potential reinforcement with matrix PLLA through the stereocomplexation. First, the surface of n-HA was silanized using a silane-coupling agent; subsequently, the ROP of D-lactide initiated by the hydroxyl group on the surface of n-HA was performed to produce an n-HA with the covalently grafted PDLA on the surface (mHA) at a high grafting rati The mHA was then compounded with PLLA (M_W_ = 1350 kg/mol) to fabricate mHA/PLA nanocomposites via solution casting. The mHA/PLA nanocomposites were comprehensively characterized in terms of fracture morphology, mechanical properties, thermal properties, crystallinity, and crystallization kinetic The in vitro cytotoxicity and osteogenic capability of the mHA/PLA nanocomposites toward rat bone marrow stromal cells (rMSCs) were also evaluated.

## 2. Materials and Methods

### 2.1. Materials

D-Lactide (DLA, 99%, Shenzhen Bright China Industrial C, Ltd., Shenzhen, China) was recrystallized from anhydrous ethyl acetate before us PLLA (weight average molar mass (M_W_) = 1350 kg/mol, Dalian Meilun Biological Technology C, Ltd., Dalian, China), calcium nitrate tetrahydrate (Ca(NO_3_)_2_·4H_2_O, 99%, Sigma-Aldrich, St. Louis, MI, USA) and diammonium phosphate ((NH_4_)_2_HPO_4_, 99%, Sigma-Aldrich, St. Louis, MI, USA) were used as received. Tin (II) 2-ethylhexanoate (Sn(Oct)_2_, Sigma-Aldrich, St. Louis, MI, USA) was re-distilled under vacuum before us Other reagents were of analytical grade and used without purification. Acicular n-HA and clavate n-HA (Nanjing EPRI nano material C, Ltd., Nanjing, China) were dried at 100 °C overnight before use, and schistose n-HA was synthesized in the lab according to the literature report (See Appendix A).

Fetal bovine serum (FBS, Gibco Life Technologies, Grand Island, NE, USA), penicillin-streptomycin (Pen/Strep, Gibco Life Technologies, Grand Island, NE, USA), bovine serum albumin (BSA, Gibco Life Technologies, Grand Island, NE, USA), Dulbecco’s modified Eagle medium (DMEM, Gibco Life Technologies, Grand Island, NE, USA), Alizarin red S (ARS, ScienCell, San Diego, CA, USA), Alkaline Phosphatase (ALP) assay kit (Beyotime, Shanghai, China), bicinchoninic acid (BCA) protein assay kit (Beyotime, Shanghai, China), paraformaldehyde (PFA, Servicebio, Wuhan, China), 10 mM β-glycerophosphate(Sigma-Aldrich, St. Louis, MI, USA), 100 nM dexamethasone (Sigma-Aldrich, St. Louis, MI, USA), Triton X-100 (Amresco, Washington, WA, USA), Osteocalcin monoclonal antibody (eBioscience, San Diego, CA, USA, 1:400 diluted in 1% BSA before use), Goat Anti-Mouse IgG H&L (Alexa Fluor^®^ 488) (Invitrogen, Chicago, IL, USA, 1:500 diluted in 1% BSA before use) and 4′,6-diamidino-2-phenylindole (DAPI, Beyotime, Shanghai, China, 1:500 diluted in 1% BSA before use) were used as received.

Rat bone marrow stromal cells (rMSCs) were isolated from the femurs and tibias of 2-week-old male Sprague–Dawley rats (Laboratory Animal Centre of Sun Yat-sen University, Guangzhou, China) following the reported protocol [43], and cultured in growth medium (GM) containing DMEM supplemented with 10% FBS and 1% Pen/strep. Osteogenic medium (OM) was prepared by adding 10% of FBS, 1% of Pen/strep, 10 mM of β-glycerophosphate, 50 μM of vitamin C and 100 nM of dexamethasone to the DMEM.

### 2.2. Surface Modification of n-HA

A total of 5.0 g of n-HA was dispersed in a 30 mL of a mixture of ethanol and DI at 10° A hydrolysate comprising 30 mL ethanol and water mixed with 1.0 g of the silane-coupling agent (3-Aminopropyltriethoxysilane (APTEOS)) was added. The pH was adjusted to 9–10 by adding 10% sodium hydroxide solution. The reaction solution was heated to 70 °C and the temperature was maintained for 8 h at a constant stirring of 400 rpm. This was followed by centrifugation to separate the product, washing with hot DI water three times, and vacuum drying at 130 °C for 8 h to yield HA@APTEOS.

Subsequently, n-HA grafted with PDLA (mHA) was synthesized via the ROP of D-lactide from the surface of n-HA, during which the hydroxyl group on the n-HA surface acted as the initiato In detail, the anhydrous HA@APTEOS and D-lactide were transferred to a silanized glass ampoule at feed ratio of 1:4; to this mixture was added Sn(Oct)_2_ (0.1 wt%) as a catalyst, and the ampoule was purged with N_2_ for three times and then sealed under vacuum (<20 Pa). The reaction was performed at 140 °C for 48 h and cooled to room temperatur The crude product was extracted with dichloromethane five times and vacuum dried at 60 °C for 48 h to yield mHA. The mHA was made from n-HA with different morphology, namely acicular n-HA(mHA_1_), clavate n-HA(mHA_2_) and schistose n-HA(mHA_3_).

### 2.3. Solution Casting of mHA_1–3_/PLA Nanocomposite Films

The mHA_1–3_/PLA nanocomposite films were fabricated via solution casting. The mHA was dispersed in dichloromethane under ultrasonication and stirred continuously for 24 h to ensure that the PDLA chains on the surface of n-HA were fully extended. PLLA (M_W_ = 1350 kg/mol) was dissolved in dichloromethane at a constant stirring rate of 600 rpm for at least 24 h to achieve a homogeneous solution. The two solutions were mixed at predetermined ratio, stirred continuously at 600 rpm for 24 h, poured into a glass dish and evaporated slowly for 72 h, after which the film was vacuum dried to constant weight. The films prepared by this method were denoted as mHA_1_/PLA, mHA_2_/PLA, and mHA_3_/PLA nanocomposites with a weight percentage of the corresponding mHA of 1.0 wt%. In addition, composites of PLLA and different pristine HA (1.0 wt% of HA_1_, HA_2_ and HA_3_, respectively) were fabricated as the control following the same protocol, and were named HA_1_/PLA, HA_2_/PLA and HA_3_/PLA, respectively.

### 2.4. Characterizations

#### 2.4.1. Scanning Electron Microscopy (SEM)

The morphologies of HA_1–3_ and the fractured surfaces of PLLA and the mHA_1–3_/PLA nanocomposites were observed under SEM (Hitachi, Regulus 8230, Hitachi, Japan) at an accelerating voltage of 10 kV.

#### 2.4.2. Fourier Transform Infrared Spectroscopy (FT-IR)

FT-IR spectra of HA, HA@APTEOS and mHA_1–3_ were recorded at wavenumbers ranging from 4000 to 400 cm^−1^ and at a resolution of 4 cm^−1^, using a TENSOR II spectrometer (Bruker, Vector-22, Karlsruhe, Germany) by the KBr method.

#### 2.4.3. X-ray Diffraction (XRD)

The phase and structural properties of HA_1–3_, PLLA and the mHA_1–3_/PLA nanocomposites were investigated via XRD (PANalytical, Empyrean, Almelo, The Netherland) in a 2*θ* angular range of 5–60° at a scan rate of 0.01°.

#### 2.4.4. Thermogravimetric Analysis (TGA)

The PDLA oligomers grafted onto the surface of HA were quantified on a SMP/PF7548/MET TGA equipment (Mettler Toledo, Zurich, Switzerland) under N_2_ atmosphere (40 mL/min), and the samples were heated from room temperature to 600 °C at a heating rate of 10 °C/min.

#### 2.4.5. Dynamic Light Scattering (DLS)

The particle size and distribution of HA_1–3_ and mHA_1–3_ were obtained through dynamic light scattering (DLS) analysis on a Zetasizer Nano ZS90 (Malvern, Malvern, UK), using dichloromethane as the solvent.

#### 2.4.6. Polarized Optical Microscopy (POM)

The crystallizing processes of the PLLA and mHA_1–3_/PLA nanocomposites were observed under a DM2500 polarized optical microscope (Leica, Weztlar, Germany) on a THM S600 hot stage (Linkam, Lincoln, UK). The samples were heated from 30 °C to 250 °C at a rate of 30 °C/min and maintained at 250 °C for 3 min. Micrographs of the spherulites were taken after a predetermined time interval during the isothermal crystallization at 120 °C.

#### 2.4.7. Tensile Testing

The tensile properties of the samples were measured according to ASTM D882-02 using an Instron 5566 Tensile Tester (Instron, Boston, MA, USA) at a speed of 4 mm/min. The films were cut into custom strip samples (width: 10 mm, total length: 70 mm and thickness: <1 mm) and the thicknesses of the samples were measured using a screw micromete Five identical samples of each type were fabricated. The interfacial shear strength formula applied was:(1)$$\tau_{i}=\frac{P_{i}}{2A}$$
where $P_{i}$ is the maximum stress below a 5% strain (N) and *A* is the cross sectional area (mm^2^).

#### 2.4.8. Differential Scanning Calorimetry (DSC)

The thermal behavior and crystallinity of the samples was determined via DSC (Netzsch, DSC-204, Bavaria, Germany). Each sample (~5 mg) was first heated from 30 °C to 240° The temperature was maintained at 240 °C for 5 min to eliminate the thermal history. Then, the samples were cooled to 30 °C at 10 °C/min and re-heated to 200 °C at 10 °C/min.

Non-isothermal crystallization test: First, the temperature was raised from room temperature to 240 °C at a heating rate of 20 °C/min. The temperature was maintained for 3 min to eliminate the heating history. Then, the samples were cooled to room temperature at a cooling rate of 10 °C/min. Finally, the temperatures of different samples were increased to 240 °C at heating rates of 1.25, 2.50, 5.00, 7.50 and 10.00 °C/min, respectively.

The glass transition temperature (*T_g_*), cold crystallization temperature (*T_cc_*), melting temperature (*T_m_*), cold crystallization enthalpy (∆*H_cc_*) and melting enthalpy (∆*H_m_*) were determined from the second heating scan. The crystallinity (*X_c_*) of neat PLLA, and the mHA_1_/PLA, mHA_2_/PLA and mHA_3_/PLA nanocomposites were calculated using:(2)$$X_{C}\left(\%\right)=\frac{∆H_{m}+∆H_{cc}}{∆H_{m}^{0}\times \left(1-\frac{wt.\%filler}{100}\right)\times 100}$$$∆H_{m}^{0}=93.0\;J/g$ is the melting enthalpy of a 100% pure crystalline sample of PLLA. *wt.%**filler* = 5% is the weight percentage of mHA_1–3_.

The relative crystallinity *X*(*T*) of the polymer was calculated using:(3)$$X\left(T\right)=\frac{X_{c}\left(T_{O}\right)}{X_{c}\left(T_{\infty }\right)}=\frac{\int_{0}^{t}\frac{dH_{c}\left(t\right)}{dt}dt}{\int_{0}^{\infty }\frac{dH_{c}\left(t\right)}{dt}dt}$$
where *T* is the crystallization temperature at time *t*, $T_{O}$ is the temperature at the start of crystallization, $T_{\infty }$ is the temperature at the end of crystallization and *dH*_c_ is the crystallization enthalpy released in the infinitely small *dT* temperature rang

Crystallization temperature (*T*) can be converted to crystallization time through the equation:(4)$$t=\frac{\left|To-T\right|}{\emptyset }$$
where ∅ is heating rate, °C/min.

The non-isothermal crystallization kinetics of polymers were analyzed using the Avrami equation:(5)$$\mathrm{lg}\left[-\ln\left(1-X\left(T\right)\right)\right]=\mathrm{lg}Z_{t}-n\mathrm{lg}t$$
where *n* is the Avrami index and $Z_{t}$ is the crystallization rate (°C/min). For non-isothermal crystallization and to obtain the crystallization rate constant at a normalized heating rate, *A*. Jeziorny proposed the correction of $Z_{t}$ to $Z_{c}$ using the formula:(6)$$\mathrm{lg}Z_{c}=\frac{\mathrm{lg}Z_{t}}{\emptyset }$$

#### 2.4.9. Cell Seeding

The PLA nancomposite films were cut into 10 × 10 mm^2^ pieces and sterilized under UV irradiation overnight in a biosafety hood, and transferred to non-stick 24-well plat rMSCs suspension in 150 μL of GM were added on the films at a seeding density of 2 × 10^4^ cells/well, and incubated for four hours to allow cell attachment, then each well was filled with 500 μL of GM. 24 h later, these films were transferred to 500 μL of GM or OM at 37 °C, 5% CO_2_, and the medium was changed every 48 h. The first time of changing the medium was defined as day 0.

#### 2.4.10. Hemolytic Property of HA

The hemolytic activity of HA, mHA_1_, mHA_2_ and mHA_3_ was evaluated. In detail, rat erythrocytes were isolated from the rat blood, followed by centrifugation at 1000 rpm for 10 min and treated with phosphate buffer (PBS; pH = 7.4) [44]. The HA/mHA dispersed in PBS was 0.0125, 0.025, 0.05 and 0.1 g/mL suspension was incubated at 37 °C for three hours, and then centrifuged at 1000 r/min for 10 min. The absorbance at 540 nm was measured on a Synergy-4 microplate reader (BioTek, Winooski, VT, USA). DI water was used as a positive control.

#### 2.4.11. Cytotoxicity of the HA/mHA

The rMSCs suspension in 100 μL of GM were seeded in 96-well plate at the density of 2 × 10^4^ cells/well, and incubated for 24 h; then, the medium was aspirated, and the wells were replenished with HA/mHA suspension in DMEM at varied concentrations (0.0125, 0.025, 0.05 and 0.1 g/mL) for 24 h. Subsequently, the HA/mHA suspension was replaced by 100 μL of GM and 10 μL of CCK-8 solution and incubated at 37 °C for two hours, and the absorbance of the upper medium at 450 nm was measured on a Synergy-4 microplate reader (BioTek, Winooski, VT, USA).

#### 2.4.12. Cell Proliferation on mHA/PLA Composites

Cells were seeded on PLA nancomposite films, as described abov At a predetermined time point (one, four and 7 days), the culture medium was aspirated, and the mHA_1_/PLA composite films were transferred to another 24-well plate to avoid the error from the cells growing on the bottom of the well; 500 μL of GM and 50 μL of CCK-8 solution were added to each well and incubated at 37 °C for two hour Then, 100 μL of the upper medium was transferred to a 96-well microplate and subjected to a measurement of the absorbance at 450 nm on a Synergy-4 microplate reader (BioTek, Winooski, VT, USA).

#### 2.4.13. Alizarin Red S (ARS) Staining and Quantification

Calcium deposition (mineralization) on the surface of the PLA nancomposite films was assayed by ARS staining. After 7, 14 and 21 days of culture, the cells were fixed with 4% PFA at 37 °C for 30 min, then stained with 500 μL of alizarin red solution (2 w/v%) for 30 min. The staining solution was aspirated and washed with PBS until no nonspecific staining could be visually observed under the light microscop For a quantitative analysis, 500 μL of cetylpyridinium chloride solution (100 mM) in PBS was added to each well to dissolve the calcium deposition; then, 100 μL of the supernatant was transferred to a 96-well plate and measured at 570 nm on a Synergy-4 microplate reader (BioTek, Winooski, VT, USA).

#### 2.4.14. ALP Activity Measurement

The ALP activity of the cells on the PLA nanocomposite films was evaluated according to the previously reported method [45] on 3, 7 and 14 days of culture; and the ALP activity was also quantified using an ALP quantification kit according to the manufacturer’s instruction.

#### 2.4.15. Osteocalcin (OCN) Immunofluorescence Staining

After being cultured in osteogenic medium for 14 days, the cells on the surface of the nanocomposite were fixed with 4% PFA solution for 30 min. Then, the samples were treated with 0.1% Triton X-100 for 5 min and blocked with 1% BSA for 30 min. Afterward, it was incubated with osteocalcin monoclonal antibody overnight at 4 °C and with Goat Anti-Mouse IgG H&L (Alexa Fluor^®^ 488) for two hour Finally, the nuclei were stained with DAPI and observed under an OLYMPUS IX71 fluorescence microscope (Olympus, Hitachi, Japan).

#### 2.4.16. Confocal Laser Scanning Microscopy (CLSM)

The morphologies of rMSCs cells cultured on PLA and mHA_1_/PLA films were observed under CLSM one day after inoculation. Before observation, the samples were washed with PBS and fixed with 4% PFA at room temperature for 30 min. After fixation, the cells were infiltrated by 0.1% Triton X-100 for 10 min and blocked with 1% BSA for 30 min; then, each sample was stained with Phalloidin-FITC for the F-actin filaments for 60 min and DAPI for the nucleus for 10 min in the dark. The samples were sealed on microscope slides and observed under a FV3000 CLSM (Olympus, Hitachi, Japan).

#### 2.4.17. Statistical Analysis

All experiments were performed with at least three independent samples, and data were expressed as mean ± standard deviation. Statistical comparisons were achieved using a one-way analysis of variance (ANOVA) or Student’s *t*-test. *p* < 0.05 was considered to be statistically significant. For all quantitative data, * refers to *p* < 0.05, ** refers to *p* < 0.01 and *** refers to *p* < 0.001.

## 3. Results and Discussion

As observed under SEM (Figure A1), the acicular n-HA exhibited a length of approximately 450 nm and a length-to-diameter ratio greater than 10:1; the clavate n-HA was found to have an average length of 200 nm and an average diameter of 35 nm; the schistose n-HA had an irregular quadrilateral shape with a width of approximately 50–100 nm and a length of approximately 150–200 nm with multilayer superposition.

The physicochemical properties of mHA_1–3_ were characterized by XRD, FTIR, TGA and DLS (Figure 1). It was found that diffraction peaks of HA_1–3_, corresponding to phase pure HA at the prominent planes (002), (211), (112), (310) and (202), etc., were obtained and matched with the JCPDS file n09-432 (Figure 1a). In the FTIR spectra (Figure 1b), the infrared spectrum of HA@APTEOS shows the characteristic peak of Si-O-Si at 1150 cm^−1^, indicating the successful grafting of the silane-coupling agent. The additional peak in the spectra of mHA_1_, mHA_2_ and mHA_3_ at 1750 cm^−1^ corresponds to the carbonyl group (C=O) and the 2900–3000 cm^−1^ corresponds to the alkyl groups (CH_3_- and -CH_2_-) of PDLA, indicating the successful grafting of D-lactid The TGA (Figure 1c) curves demonstrated that the decomposition of PDLA occurred at approximately 250 °C, and HA was stable upon heating in N_2_. During the whole heating process, the mass loss of HA@APTEOS is 7.2%. Therefore, the grafting amount of PDLA onto HA_1–3_ was able to be calculated from those weight losses of mHA_1–3_ in subtracting the weight loss of HA@APTEOS, which were found to be 24.8 wt%, 22.3 wt% and 21.9 wt%, respectively, for mHA_1_, mHA_2_ and mHA_3_.

The average particle sizes of HA_1_, HA_2_ and HA_3_, as determined by the DLS method, were 400 nm, 350 nm and 800 nm, respectively. In contrast, the average particle sizes of the mHA_1_, mHA_2_ and mHA_3_ samples were 240 nm, 160 nm and 700 nm, respectively, displaying a smaller particle size and narrower particle size distribution than those of the HA_1–3_ (Figure 1d–f). This indicates that the surface-grafted PDLA prevented HA from aggregating in chloroform, thereby improving its dispersibility.

The suspensions were allowed to stand for 12 h after sonication; it was found that the HA_1–3_ suspensions displayed poor stability in chloroform, and the HA_1–3_ particles slowly settled at the bottom of the ampoules (Figure 2a), with no significant precipitate observed in the mHA_1–3_ samples (Figure 2b). It was hypothesized that the PDLA chains on the surface of mHA_1–3_ swell and extend in chloroform, thereby preventing mHA_1–3_ from re-aggregating in the solvated stat Upon the mixing of mHA_1__-3_ suspensions with PLLA solution, interactions between the PDLA chains on the mHA_1–3_ surface and the swollen PLLA chains may occur, significantly improving the stability of the suspensions and inhibited the precipitation of mHA_1–3_ in the mixture solution. This is critical for the fabrication of homogeneous mHA_1–3_/PLA nanocomposite films via the solution-casting technique, after which the mHA_1–3_ may be more evenly distributed in the PLLA matrix.

The mechanical performance of the mHA_1–3_/PLA nanocomposites was evaluated in terms of tensile strength, elongation-at-break, Young’s modulus and interfacial shear strength, respectively (Figure 3). It was found that tensile strength of neat PLLA was approximately 67 MPa. After 1% of HA_1_ was added, the tensile strength increased slightly by approximately 2 MPa. This may be due to the significant agglomeration of HA_1_ in the matrix. However, the tensile strength increased from 67 to 79 MPa with 1% of mHA_1_. The tensile strength gradually increased upon the content of mHA_1_, and a maximum tensile strength of 91 MPa was achieved with 5% of mHA; as the content of mHA_1_ further increased to 10%, the tensile strength decreased because the high content of mHA_1_ in the PLLA matrix may possibly affect the distribution of filler and develop defects, resulting in performance degradation. The elongation-at-break and tensile strength displayed similar trends, that is, when the content of mHA_1_ was 5%, the elongation-at-break peaked at 11.6%, which was higher than of the neat PLLA with a value of 6.5% (Figure 3a). At the content of mHA_1_ lower than 5%, the Young’s modulus and the interfacial shear strength of mHA_1_ (5%)/PLA nanocomposite increased by 57% and 26%, respectively (Figure 3b). According to the shear yield crazing theory [46,47], when mHA_1_ was homogeneously dispersed in the PLA matrix, the stress concentration induced significant crazing, which absorbed a large amount of energy on impact. Moreover, the excellent interfacial interaction between mHA_1_ and PLLA can help control the propagation of crazing and prevent destructive cracking.

In the mHA_2_/PLA nanocomposites, when increasing the content of mHA_2_ to 3%, the tensile strength increased from 67 to 90 MPa; further increasing the content of mHA_2_ to 5% did not result in significant improvement of the tensile strength or elongation-at-break. The maximum Young’s modulus, elongation-at-break and interfacial shear strength increased by 55%, 64% and 10%, respectively, as compared with those of the neat PLLA (Figure 3c,d).

The mHA_3_/PLA nanocomposite displayed a different reinforcing performance (Figure 3e,f). It was found that at the content of mHA_3_ of 3%, this tensile strength was 31% higher than that of PLA, which was also the highest value in the studied rang The Young’s modulus, interfacial shear strength, and elongation-at-break increased by 50%, 21% and 60%, respectively.

The improvement in mechanical properties at the interface between mHA_1–3_ and PLLA matrix may be attributed to the promotion of stress transfer by the uniformly dispersed high-modulus mHA_1–3_, which results in bifurcation and the deflection of micro-cracks, consequently facilitating energy dissipation. More importantly, due to the presence of intermolecular stereocomplexation, mHA_1–3_ can play a bridging role to prevent the propagation of micro-cracks at the interface. The mHA_1–3_ samples displayed different degrees of enhancement, which may be related to their degree of dispersion and siz

The fractured surfaces of mHA_1–3_/PLA nanocomposites were observed under SEM (Figure 4). It was found that the neat PLLA displayed a typical single-phase morphology with a smooth surface, indicating a brittle fracture (Figure 4a,a’). In the mHA_1–3_/PLA nanocomposites, the rough surface with ductile dimples and tear edges was observed. This indicates that the addition of mHA_1–3_ caused the fractures of PLLA to change from brittle to ductil No noticeabe difference in the fractured surfaces of the mHA_1_/PLA, mHA_2_/PLA and mHA_3_/PLA nanocomposites was observed, although all of them were coarser than that of the neat PLLA.

The influence of mHA_1_, mHA_2_ and mHA_3_ on the crystallization behavior of the PLA matrix was studied (Figure 5). In the neat PLLA, a few spherulites appeared after 2 min and the number of spherulites increased slightly in the following 2 min (Figure 5a,a’). In contrast, numerous spherulites formed in the mHA_1–3_/PLA nanocomposites after 2 min and the spherulites covered almost the entire field of view after 2 min. After the same time interval, the spherulites in the nanocomposites were more numerous and smalle Compared with mHA_3_/PLA, more spherulites appeared in the mHA_1_/PLA and mHA_2_/PLA nanocomposites after 2 min, with little difference in between them (Figure 5b–d). At 4 min, the spherulites of all the composites filled the entire field of view (Figure 5b’–d’). The improved crystallization of the PLLA matrix due to the addition of mHA_1–3_ could be mainly attributed to the synergistic effect of mHA_1–3_ as the nucleating agent and stereocomplex crystals are nucleating site This synergistic effect promoted a higher nucleation density and a smaller spherulite size in the composites, which increased the crystallization rate [48].

The crystallization behavior was further studied using DSC (Figure 6 and Table 1). In the first cooling curves, the intensity of the cooling crystallization peaks of the three nanocomposites was significantly higher than that of the neat PLLA (Figure 6a). In the second heating curves, the cold crystallization peaks of the mHA_1–3_/PLA nanocomposites shifted to lower temperatures, indicating that mHA_1–3_ as a nucleating agent promoted the crystallization of the nanocomposit Comparing the curves of the mHA_1_/PLA, mHA_2_/PLA and mHA_3_/PLA nanocomposites, it is clear that the cold crystallization peak of mHA_1_/PLA was sharper than that of mHA_2_/PLA and mHA_3_/PLA, indicating that mHA_1_ was more conducive to the crystallization of PLA (Figure 6b).

Compared with neat PLLA, the crystallinities of mHA_1_/PLA, mHA_2_/PLA and mHA_3_/PLA increased by 92.7%, 100.0% and 89.3%, respectively. The ability of mHA_1–3_ to promote crystallization could be attributed to the synergistic effect of mHA_1–3_ as a nucleating agent and the stereocomplex crystals as nucleating centers [49].

The crystal patterns of neat PLLA and the mHA_1–3_/PLA nanocomposites were characterized (Figure 7). The diffraction peaks for neat PLLA at 2*θ* = 16.8° and 19.0° were attributed to the characteristic peaks of the *β* crystal corresponding to the (200/110) and (203) crystal planes, respectively. The peak intensities indicate that the obtained samples have good crystallinity. For the mHA_1–3_/PLA nanocomposites, the characteristic peaks of the *β* crystals and the stereocomplex (*sc*) crystals existed simultaneously. The diffraction peaks at 2*θ* = 11.6°, 20.8°, and 23.9° were attributed to *sc* crystallites and corresponded to the (110), (300)/(030), and (220) lattice planes, respectively. The formation of *sc* crystals was due to the stereocomplex interactions between the PDLA grafted onto the surface of mHA_1–3_ and the PLLA matrix. The *sc* crystals simultaneously acted as nucleation sites to promote the crystallization of PLLA.

Because crystallization has a significant influence on the mechanical properties of materials, the kinetics of the non-isothermal crystallization of the nanocomposites were investigated at varied heating rates (Figure 8). With an increasing heating rate, the peak crystallization temperatures (*T_p_*) of all the materials shifted toward higher temperatures and the exothermic peaks became wide This was due to the gradual increase in the formation of imperfect crystals during crystallization [50,51]. At the same heating rate, the *T_p_* values of the mHA_1–3_/PLA nanocomposites were lower than that of the neat PLLA, indicating that the addition of mHA_1–3_ effectively promoted the cold crystallization nucleation of the PLLA matrix.

The relative crystallinities of PLLA and the mHA_1–3_/PLA nanocomposites were plotted against the temperature at different heating rates (Figure 9). All the curves were S-shaped, indicating that the crystallization of the materials could be divided into three stages: the slow nucleation stage, the fast primary nucleation stage and the slow secondary nucleation stag Moreover, all the curves shifted to higher temperatures at a higher heating rate, indicating that the acceleration of the heating rate makes the samples not have enough time to form the crystal nuclei.

The relative crystallinity versus time at different heating rates was also investigated (Figure 10). The curves gradually shifted to the left at higher heating rate The half crystallization times (*t*_1/2_) values of PLA and the mHA_1–3_/PLA nanocomposites decreased with the increasing heating rate (Table 2). Moreover, at the same heating rate, the *t*_1/2_ values of the mHA_1–3_/PLA nanocomposites were lower than that of PLA, indicating that mHA_1–3_ can promote heterogeneous nucleation and improve the crystallization of the PLA main chain.

The crystallization parameters were calculated using (5) and (6) (Table 3). n_1_ and Z_c1_ are the parameters of the primary crystallization stage, while n_2_ and Z_c2_ are the parameters of the secondary crystallization stag The n_1_ value range of neat PLLA was 3.61–6.93 and the n_1_ value range of the mHA_1–3_/PLA nanocomposites was 3.20–8.49, indicating that the crystal growth mode involved simultaneous two-dimensional and three-dimensional growth.

In addition, the n_1_ values of all samples increased with the increasing heating rate and the n_2_ values were lower than the n_1_ value This was due to a change in the growth mode of the crystals during the secondary crystallization stag Secondary crystallization involves the perfection of crystals, including the elimination of defects and the melting/recrystallization of small crystal The Z_c1_ and Z_c2_ values of PLLA and the mHA_1–3_/PLA nanocomposites decreased with increasing heating rate and the Z_c1_ and Z_c2_ values of the nanocomposites were larger than those of the PLLA. These results confirm that the addition of mHA_1–3_ improved the crystallization rate of the PLLA matrix.

The mHA_1_/PLA and mHA_2_/PLA nanocomposites exhibited similar crystallization kinetic This was attributed to the minor difference in morphology and size between mHA_1_ and mHA_2_. Therefore, their kinetic effects on the crystallization behavior of the matrix were simila However, the mHA_3_/PLA nanocomposite displayed a different crystallization behavio This was attributed the irregular, quadrilateral shape of mHA_3_, which was also multilayered, leading to a stronger anisotropic effect. By reducing the specific surface area, mHA_3_ plays a different role from mHA_1_ and mHA_2_ due to its unique propertie

The in vitro biocompatibility of mHA_1–3_ was evaluated briefly in terms of hemolytic ratio and cytotoxicity (Figure 11). The red blood cells in the DI water group were found almost completely ruptured, and those in other groups were deposited at the bottom of the centrifuge tube (Figure 11b). No significant difference between the mHA_1–3_ group and the PBS group could be observed, indicating that the red blood cells were intact. Quantitative analysis evidenced hemolysis ratios lower than 5% in the case of nanoparticles at different concentrations, indicating that the mHA_1–3_ does not cause obvious hemolysis and has appreciable hemocompatibility.

The cytotoxicity of materials is of primary importance (Figure 11c). In this case, it was found that the cell survival rate decreased slightly at higher concentration of nanoparticles; however, it was still higher than 80% at the concentration of nanoparticles of 0.1 g/mL, suggesting an acceptable cytotoxicity of these nanoparticles at high concentration. It was also observed that there were no significant difference between different groups, which further proved the cytocompatibility of the mHA_1–3_ nanoparticle

The proliferation of rMSCs, cultured on the surface of mHA_1_/PLA films with varied concentrations of mHA_1_, was measured as a function of culture time (Figure 11d). In general, the cells in each group proliferated significantly with tim Compared with these on the surface of the neat PLA film, the rMSCs cultured on mHA_1_/PLA film demonstrated better cell proliferation, and the mHA_1_ (10%)/PLA group demonstrated higher proliferation when the cells were cultured at day 7. The above results indicated that the nanocomposite films have better performance in supporting the proliferation of rMSC

The calcium nodules formed by rMSCs on the mHA_1_/PLA films were visualized through ARS staining, and the calcium deposition was quantified (Figure 12a–c). It was found that when the rMSCs were cultured for 7, 14 and 21 days, red spots appeared in the ARS-stained samples and a clear color progression gradient with culture time was observed. Compared with that on the neat PLA, more red spots on the surface of the nanocomposite films could be observed (Figure 12a,b); quantitative analysis also demonstrated a significantly increased calcium deposition on the nanocomposite films, suggesting that the nanocomposite can promote mineralization of rMSCs (Figure 12c). The mHA_1_ (5%)/PLA and mHA_1_ (10%)/PLA nanocomposite films exhibited significantly higher calcium deposition levels than other film These results demonstrate that high concentration of mHA_1_ can promote the mineralization of rMSC

The ALP activity of rMSCs on the surface of mHA_1_/PLA was assayed (Figure 12d). Higher level of ALP activity was achieved on the mHA_1_ (10%)/PLA film than the other groups after 7 days of cultur After 14 days of culture, the mHA_1_/PLA composite group demonstrated elevated ALP activity compared with the PLA group, and the mHA_1_ (10%)/PLA group demonstrated the highest ALP activity among these group Thus, rMSCs grown on nanocomposite films containing mHA_1_ would yield higher levels of ALP expression.

The expression of OCN was evidenced by immunofluorescence staining (Figure 12e). A stronger fluorescence intensity (green fluorescence) was measured for mHA_1_/PLA nanocomposite films compared with that of PLA. In addition, a significant increase in fluorescence intensity was observed with increasing mHA_1_ content, suggesting that the presence of mHA_1_ may promote the expression of OCN in rMSC

The spreading of rMSCs on neat PLA and mHA_1_/PLA nanocomposite films was observed under CLSM (Figure 13). Although the cells spread on the surface of neat PLA, the spreading area of the cells was quite limited, and actin stress fibers were poorly developed. The F-actin plays a crucial role in cell spreading, and there is a direct correlation between poor cell spreading and low cell viability on the surface of PLA. Significantly better cell spreading and higher amounts of actin filaments on the surface of mHA_1_/PLA films were observed compared with those on the neat PLA film. A higher content of mHA_1_ seemed to favor the cell spreading and adhesion. These findings, in addition to the above osteogenic differentiation assays, clearly suggest that the mHA/PLA nanocomposites can be used as excellent candidate materials in potential bone repair application

## 4. Conclusions

In this study, stereocomplexation-reinforced high-strength mHA/PLA composite is prepared to investigate its mechanical properties and bone repair potential. In detail, the addition of mHA increased the tensile strength, the Young’s modulus, the interfacial shear strength, the elongation-at-break and the crystallinity of PLA by 34%, 53%, 26%, 70% and 17%, respectively. Benefiting from the stereocomplexation and the structural similarity between PDLA and PLLA, enhanced the interfacial interaction between n-HA and PLLA as well as improved the dispersion of n-HA in the PLLA matrix were achieved through surface grafting of PDLA; in addition, the stereocomplex crystals formed by the stereocomplexation of PDLA and PLLA also acted as the nucleating agents to facilitate the crystallization of the PLLA matrix. The improved dispersion of the nanofiller, enhanced interfacial interaction between the nano-filler and matrix and the elevated crystallinity of the matrix synergistically contribute to the greatly increased mechanical properties of the PLLA nanocomposit A study of crystallization kinetics demonstrates that mHA_1_/PLA and mHA_2_/PLA nanocomposites display similar crystallization kinetics and that the addition of mHA significantly increases the crystallization rate of the matrix. An in vitro cytotoxicity assay and the osteogenic differentiation of rMSCs revealed that the mHA_1_/PLA composite is cytocompatible, and can effectively induce apatite precipitation and support the adhesion, proliferation and osteogenic differentiation of rMSC These mHA/PLA nanocomposites can be used as excellent candidate materials in potential bone repair application.

## Figures and Tables

**Figure 1 polymers-14-00645-f001:**
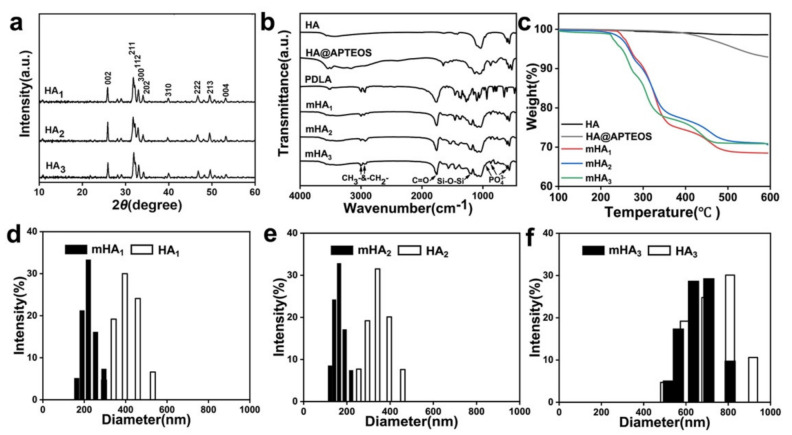
The basic physicochemical characterizations: (**a**) XRD pattern; (**b**) FTIR spectrum; and (**c**) TGA curv Particle size distributions of HA_1–3_ and mHA_1–3_: (**d**) HA_1_ and mHA_1_; (**e**) HA_2_ and mHA_2_; (**f**) HA_3_ and mHA_3_.

**Figure 2 polymers-14-00645-f002:**
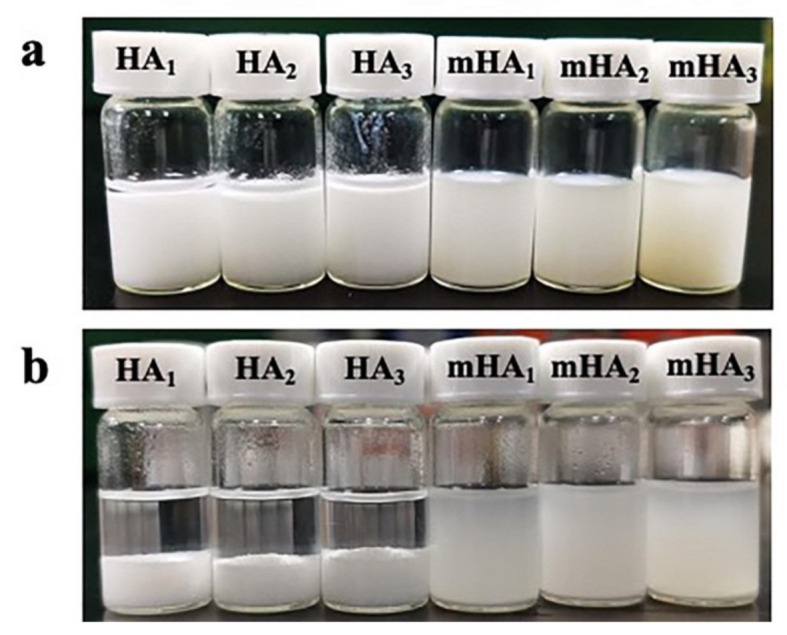
Stability of HA_1–3_ and mHA_1–3_ in chloroform. (**a**) Digital photos after 0 h and (**b**) after 12 h.

**Figure 3 polymers-14-00645-f003:**
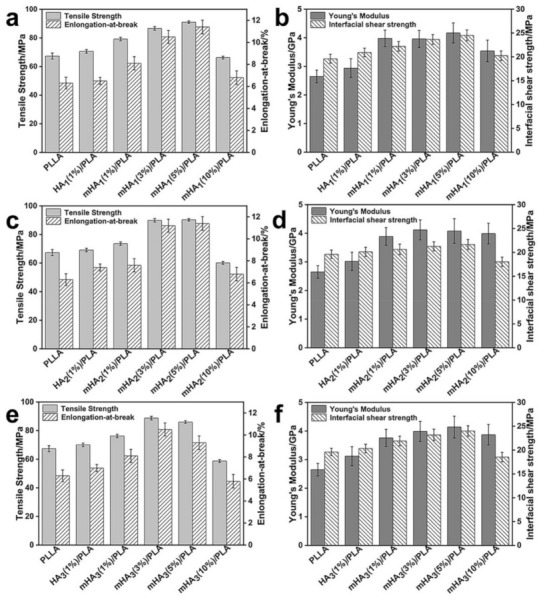
Mechanical properties of the mHA_1–3_/PLA composites: (**a**,**b**), mHA_1_/PLA; (**c**,**d**), mHA_2_/PLA; (**e**,**f**), mHA_3_/PLA.

**Figure 4 polymers-14-00645-f004:**
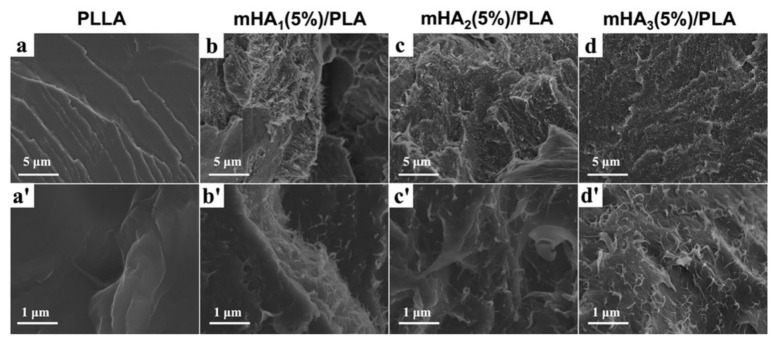
Fractured surfaces of mHA_1–3_/PLA nanocomposites: (**a**,**a’**), PLLA; (**b**,**b’**), mHA_1_/PLA; (**c**,**c’**), mHA_2_/PLA; (**d**,**d’**), mHA_3_/PLA.

**Figure 5 polymers-14-00645-f005:**
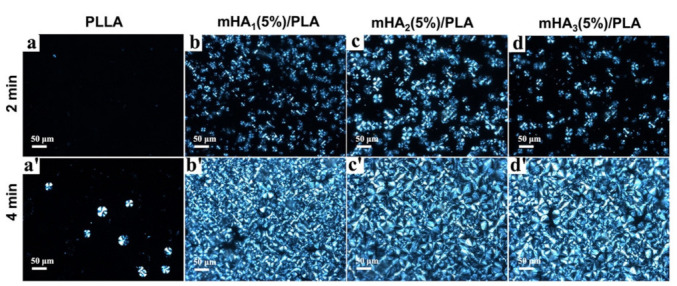
Polarized micrographs of mHA_1–3_/PLA nanocomposites: (**a/a’**) PLLA; (**b/b’**) mHA_1_/PLA; (**c/c’**) mHA_2_/PLA; (**d/d’**) mHA_3_/PLA.

**Figure 6 polymers-14-00645-f006:**
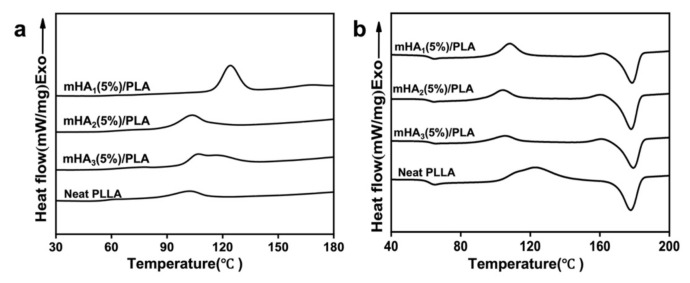
DSC curves of mHA_1–3_/PLA nanocomposites: (**a**) first cooling scan; (**b**) second heating scan.

**Figure 7 polymers-14-00645-f007:**
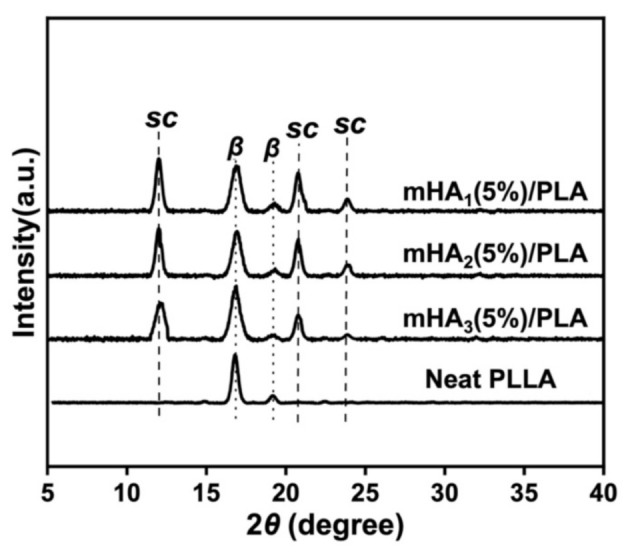
XRD spectra of PLLA and the mHA_1–3_/PLA composite.

**Figure 8 polymers-14-00645-f008:**
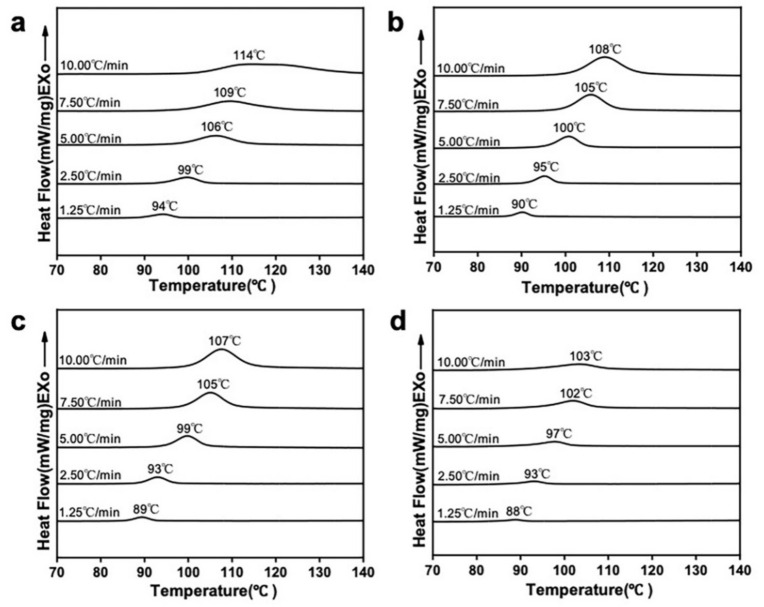
Heat flux curves of PLLA and the mHA_1–3_/PLA composites at various temperatures: (**a**) PLLA; (**b**) mHA_1_(5%)/PLA; (**c**) mHA_2_(5%)/PLA; (**d**) mHA_3_(5%)/PLA.

**Figure 9 polymers-14-00645-f009:**
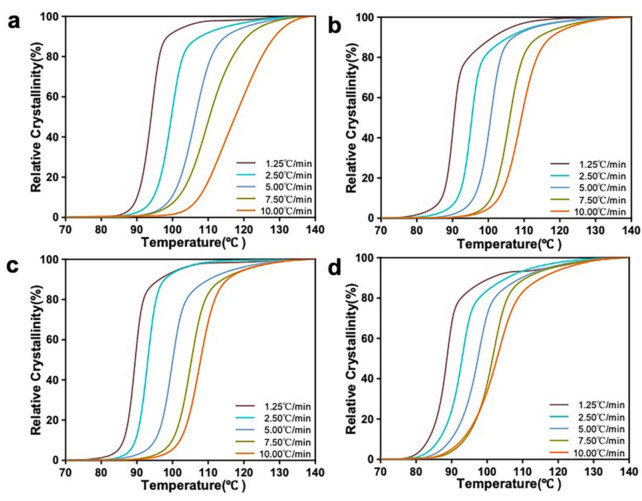
Relative crystallinity curves of PLLA and the mHA_1–3_/PLA composites at various temperatures: (**a**) PLLA; (**b**) mHA_1_(5%)/PLA; (**c**) mHA_2_(5%)/PLA; (**d**) mHA_3_(5%)/PLA.

**Figure 10 polymers-14-00645-f010:**
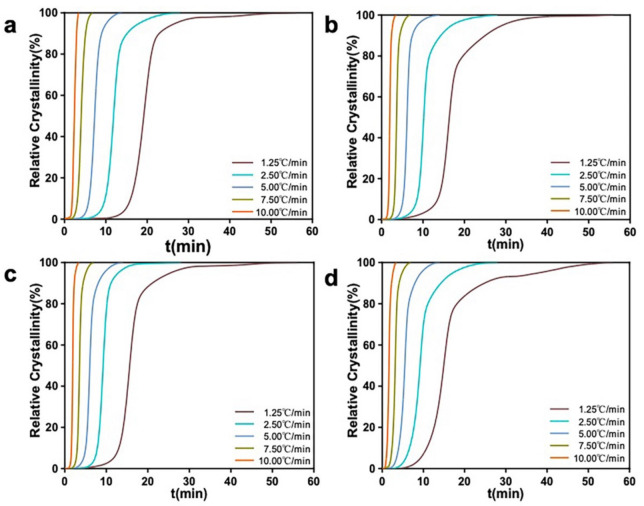
Relative crystallinity curves of PLLA and the mHA_1–3_/PLA composites over time: (**a**) PLLA; (**b**) mHA_1_ (5%)/PLA; (**c**) mHA_2_ (5%)/PLA; (**d**) mHA_3_ (5%)/PLA.

**Figure 11 polymers-14-00645-f011:**
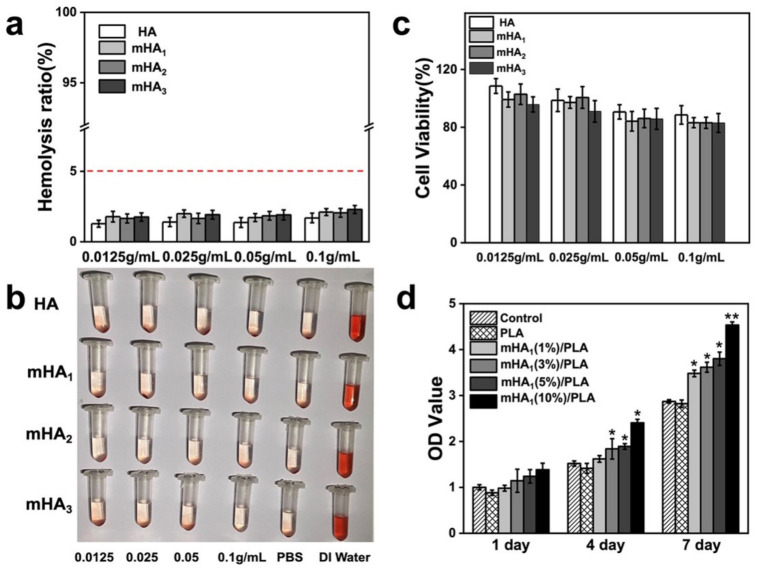
Biocompatibility of nanoparticles and nanocomposites: (**a**,**b**), hemolysis test; (**c**) cytotoxicity test; (**d**) the proliferation of rMSCs on composite surfac * refers to *p* < 0.05, ** refers to *p* < 0.01.

**Figure 12 polymers-14-00645-f012:**
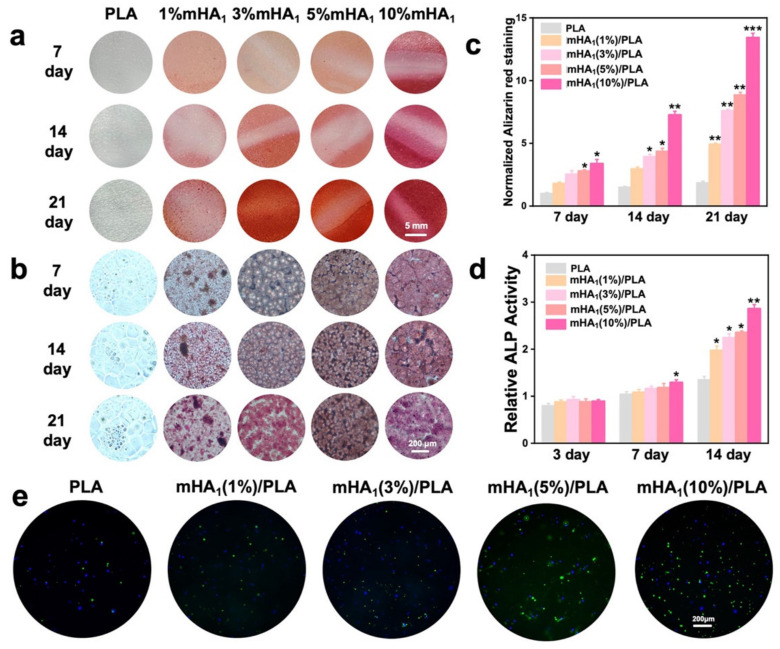
The expression of osteogenic markers of rMSCs on the surface of the composite: (**a**) AR-staining digital photos; (**b**) AR-staining light microscopy; (**c**) quantitative analysis by AR staining; (**d**) ALP activity; (**e**) the expression of osteocalcin (OCN: green fluorescence, nucleus: blue fluorescence). * refers to *p* < 0.05, ** refers to *p* < 0.01 and *** refers to *p* < 0.001.

**Figure 13 polymers-14-00645-f013:**
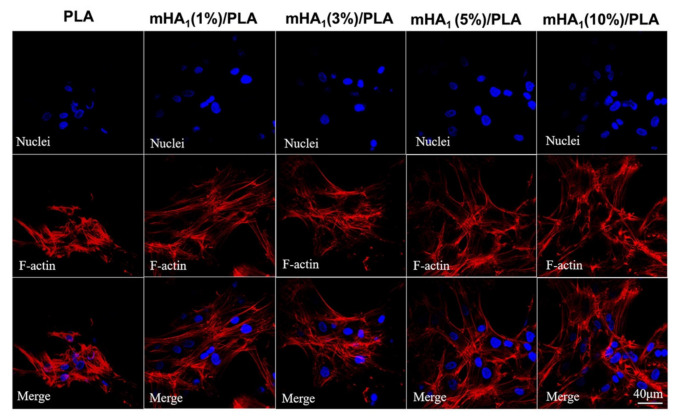
Spreading of rMSCs on surface of nanocomposite film.

**Table 1 polymers-14-00645-t001:** Thermal properties and crystallinities of PLLA and the HA_1–3_/PLA nanocomposite.

Samples	*Tg* (°C)	*Tcc* (°C)	*Tm* (°C)	**Δ*Hm* (J/g)**	**Δ*Hcc* (J/g)**	*Xc* (%)
Neat PLLA mHA_1_/PLA	58.3 63.0	122.8 108.7	178.4 179.6	37.6 44.5	−21.1 −13.0	17.8 34.3
mHA_2_/PLA	61.8	105.4	179.3	45.3	−12.6	35.6
mHA_3_/PLA	62.6	108.2	180.1	42.4	−11.4	33.7

*Tg* = glass transition temperature; *Tcc* = cold crystallization temperature; *Tm* = melting temperature; Δ*Hm* = melting enthalpy; Δ*Hcc* = cold crystallization enthalpy; *Xc* = degree of crystallinity.

**Table 2 polymers-14-00645-t002:** Semi-crystallization times of PLLA and the HA_1–3_/PLA nanocomposites at different heating rate.

Samples	1.25 °C/min	2.50 °C/min	5.00 °C/min	7.50 °C/min	10.00 °C/min
PLLA	19.16	11.87	7.31	4.08	2.39
mHA_1_/PLA	16.43	10.14	6.16	3.56	2.00
mHA_2_/PLA	15.62	9.27	5.98	3.48	1.87
mHA_3_/PLA	14.99	9.15	5.51	3.15	1.64

*t*_1/2_: Semi-crystallization time, min.

**Table 3 polymers-14-00645-t003:** Avrami indices (n) and crystallization rate constants (K(T)) of PLLA and the mHA_1–3_/PLA nanocomposites at different heating rate.

Samples	Avrami	1.25 °C/min	2.50 °C/min	5.00 °C/min	7.50 °C/min	10.00 °C/min
Neat PLLA	n_1_	3.61	4.44	5.43	6.65	6.93
	Z_c1_	4.29	4.25	4.05	3.25	1.8
	n_2_	1.15	2.06	2.78	3.27	4.24
	Z_c2_	1.19	1.12	0.98	0.84	0.57
mHA_1_/PLA	n_1_	3.41	3.96	4.66	6.35	8.17
	Z_c1_	4.54	4.42	4.2	3.94	2.62
	n_2_	1.42	1.69	1.9	2.52	3.15
	Z_c2_	1.62	1.6	1.31	1.29	0.85
mHA_2_/PLA	n_1_	3.2	3.93	4.77	6.29	8.49
	Z_c1_	4.42	4.5	4.16	3.85	2.6
	n_2_	1.13	1.29	2.08	2.52	2.85
	Z_c2_	1.23	0.95	1.54	1.29	0.68
mHA_3_/PLA	n_1_	3.59	4.43	4.14	4.98	5.38
	Z_c1_	4.62	4.55	3.38	2.76	1.31
	n_2_	1.11	1.7	1.93	2.14	2.5
	Z_c2_	1.21	1.6	1.38	0.98	0.53

Z_c1_: primary crystallization rate constant; Z_c2_: secondary crystallization rate constant; n_1_: Avrami index of primary crystallization; n_2_: Avrami index of secondary crystallization.

## Data Availability

Not applicable.

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
