# Peer review of "Stereocomplexation Reinforced High Strength Poly(L-lactide)/Nanohydroxyapatite Composites for Potential Bone Repair Applications"

_polymers, 2022, doi:10.3390/polym14030645_

Round 1

Reviewer 1 Report

Reviewer report: Stereocomplexation reinforced high strength poly(L-lac- 2 tide)/nanohydroxyapatite composites for potential bone repair 3 applications

General comment: I like these topics very much, especially in composites of nano-hydroxyapatite and other biocompatible compounds for bone regeneration. I must emphasize that there are a lot of such works, but every novelty, even a small one in this area, can contribute to the improvement of such materials.

The Abstract section is very well written.

Line 33: At the end of a sentence, it should be some reference since applications of these materials are pretty known so far. The last sentence in the paragraph needs to be rephrased; it is unclear for readers.

Line 38: Please add references regarded as HA, β-TCP and other Ca-phosphate material. Also, it is highly questionable to use calcium carbonate as a replacement biocompatible material filler.

Line 46: And also should be emphasized in the text that HA and other biocompatible materials encourage the onset of crystallization and the formation or remineralization of bones, simplified since our bones are low-crystalline carbonate hydroxyapatite osteoblast cells begin regeneration by adding implant materials in this case, HA and other calcium phosphates.

Lines 68-75: Please use shorter and concise sentences.

Lines 102-129: Are written proper.

Line 145: I suggest that authors put labels of samples in brackets, for example, acicular (mHA1), clavate (mHA2)...etc.

Line 156-160: The same comment as I suggested above, you have a lot of samples, perhaps to put in Table to be more apparent for readers.

Line 163: SEM, are samples prepared with Au coating or C coating before analysis? If you used some other method, please explain.

Line 171: Yes, the XRD method gives you information about the structure. But it is more precise to write: The phase and structural properties are investigated using the XRD method; you cannot see the structure from XRD measurements. Please add the crystallographical database and ICDD or JCPDS card used for phase identification.

Line 200-300: These parts are well written. It is a lot of methods in text. I guess this part can be shortened a bit.

Figure 1: XRD-if you put the card you used for identification, there is no need to insert the HA card in Figure a. It is unclear for readers, and there are many results in Figure 1.

Line 307: Please rephrase, it is very unclear to say typical crystalline patterns. You have diffraction peaks and identified reflections: 002, 211,112,202,310..etc. which corresponds to HA. Also, you achieved to synthesize nanocrystalline HA material according to XRD results, you have wider peaks with a slightly lower intensity that indicate nanocrystalline material. Also, it is important to note that no secondary phase was identified based on XRD results. Synthesized HA1, HA2 and HA3 have a proper structure, but it is not highly crystalline. All of these conclusions can be conducted from XRD results.

Lines 344-346: Please compare tensile strength results with literature data. Line 345: Average particle sizes for HA1 are 400 nm and for mHA1 are 240 nm it is average values compared to the other two samples. So the agglomeration is not so significant, the best tensile results for this sample may be because of agglomeration but also there are maybe other characteristics that contribute to tensile results?

Line 385: It is very uncommon to say wire-drawing. It is preferred for HA nanocrystals to have needle-like morphology and to have elongated hexagonal prismatic forms of crystals.

Line 401-402: I see from figures that inter crystal space is filled with PLA matrix, the crystals are not broken they are perhaps covered in some parts with matrix filling.

Line 413: Yes, the HA is a nucleating agent, but also with higher temperature, the structure tends to proper order.

Note: Theta is always italic, and also beta!

Figure 7: Please note what on diffractogram means SC, and beta? If you mention reflections in manuscripts 200,110 and 203 please put them in Figure. There are abbreviations in Figure 7 that are not explained in the manuscript please align text with Figure 7.

Line 430-431: The peak intensities indicate that obtained samples have good crystallinity. Please delete the sentence “The intensities of the characteristic peaks displayed a 430 positive relationship with the crystallinity” and rephrase it.

Line 445-446: Please provide a reference for these statements.

Line 509-514: Congrats! It is an excellent result!

In Conclusion, the text should be concluded from the very beginning and put some sentences about good results of obtained samples and then about characterization. The rest of the conclusion should emphasize the novelty of research once again and also put the best results from this study.

Author Response

RESPONSES TO THE REVIEWERS’ COMMENTS

Thank you for your help with our manuscript entitled “Stereocomplexation reinforced high strength poly(L-lactide)/nanohydroxyapatite composites for potential bone repair applications” (manuscript ID: 1570312). We have carefully read the valuable comments, and revised the manuscript accordingly. Please kindly check enclosed revised manuscript, in which corrections/revisions are presented in the Track Change mode. The responses to the reviewers’ comments are listed below for your evaluation.

With regards,

Chao Zhang

Reviewer #1:

COMMENT 1: Line 33: At the end of a sentence, it should be some reference since applications of these materials are pretty known so far. The last sentence in the paragraph needs to be rephrased; it is unclear for readers.

REPLY: The authors thank the reviewer for the helpful suggestion. We have accordingly revised the manuscript and the revisions are highlighted in the revised manuscript, please kindly check it for the major changes made to the manuscript (Line 33-40).

COMMENT 2: Line 38: Please add references regarded as HA, β-TCP and other Ca-phosphate material. Also, it is highly questionable to use calcium carbonate as a replacement biocompatible material filler.

REPLY: Thanks for the suggestion. We have accordingly updated the reference list.

COMMENT 3: Line 46: And also should be emphasized in the text that HA and other biocompatible materials encourage the onset of crystallization and the formation or remineralization of bones, simplified since our bones are low-crystalline carbonate hydroxyapatite osteoblast cells begin regeneration by adding implant materials in this case, HA and other calcium phosphates.

REPLY: Thanks for the suggestion. We have accordingly revised the manuscript, please kindly check for the major changes made to the manuscript (Line 55-57).

COMMENT 4: Lines 68-75: Please use shorter and concise sentences.

REPLY: Thanks for the suggestion. We have re-written this part as per the suggestion (Line 68-71).

COMMENT 5: Line 145: I suggest that authors put labels of samples in brackets, for example, acicular (mHA1), clavate (mHA2)...etc.

REPLY: Thanks for the helpful suggestion. We have accordingly changed the presentation of the sample labels (Line 171-172).

COMMENT 6: Line 156-160: The same comment as I suggested above, you have a lot of samples, perhaps to put in Table to be more apparent for readers.

REPLY: Thanks for the comment. Since there are already three tables and 13 figures in the main text, we feel it may be better to keep the tables and figures as less as possible.

COMMENT 7: Line 171: Yes, the XRD method gives you information about the structure. But it is more precise to write: The phase and structural properties are investigated using the XRD method; you cannot see the structure from XRD measurements. Please add the crystallographical database and ICDD or JCPDS card used for phase identification.

REPLY: Thanks for the helpful suggestion. We have re-written this part according to the Reviewer’s suggestion (Line 198). The JCPDS card has been added on Line 337.

COMMENT 8: Line 200-300: These parts are well written. It is a lot of methods in text. I guess this part can be shortened a bit.

REPLY: The authors thank the reviewer for the helpful suggestion. We have tried to rephrase this part.

COMMENT 9: Figure 1: XRD-if you put the card you used for identification, there is no need to insert the HA card in Figure a. It is unclear for readers, and there are many results in Figure 1.

REPLY: Thanks for the suggestion. We have updated Figure 1 according to the Reviewer's suggestion.

COMMENT 10: Line 307: Please rephrase, it is very unclear to say typical crystalline patterns. You have diffraction peaks and identified reflections: 002, 211,112,202,310..etc. which corresponds to HA. Also, you achieved to synthesize nanocrystalline HA material according to XRD results, you have wider peaks with a slightly lower intensity that indicate nanocrystalline material. Also, it is important to note that no secondary phase was identified based on XRD results. Synthesized HA1, HA2 and HA3 have a proper structure, but it is not highly crystalline. All of these conclusions can be conducted from XRD results.

REPLY: Thanks for the helpful suggestion. We have re-written this part according to the Reviewer’s suggestion (Line 335-336).

COMMENT 11: Line 385: It is very uncommon to say wire-drawing. It is preferred for HA nanocrystals to have needle-like morphology and to have elongated hexagonal prismatic forms of crystals.

REPLY: Thanks for the comment. We have re-written this sentence according to the Reviewer’s suggestion (Line 417).

COMMENT 12: Line 401-402: I see from figures that inter crystal space is filled with PLA matrix, the crystals are not broken they are perhaps covered in some parts with matrix filling.

REPLY: Thanks for the comment. Corresponding illustration has been corrected (Line 435).

COMMENT 13: Line 413: Yes, the HA is a nucleating agent, but also with higher temperature, the structure tends to proper order.

REPLY: Thanks for the comment. We agree with the reviewer.

COMMENT 14: Line 430-431: The peak intensities indicate that obtained samples have good crystallinity. Please delete the sentence “The intensities of the characteristic peaks displayed a 430 positive relationship with the crystallinity” and rephrase it.

REPLY: Thanks for the suggestion. This sentence was rephrased according to the Reviewer’s suggestion (Line 465-466).

COMMENT 15: Line 445-446: Please provide a reference for these statements.

REPLY: Thanks for the suggestion. We have accordingly updated the reference list as per the suggestion.

COMMENT 16: Line 163: SEM, are samples prepared with Au coating or C coating before analysis? If you used some other method, please explain.

REPLY: Thanks for the question. Samples were sputter-coated with Au before analysis.

COMMENT 17: Lines 344-346: Please compare tensile strength results with literature data. Line 345: Average particle sizes for HA1 are 400 nm and for mHA1 are 240 nm it is average values compared to the other two samples. So the agglomeration is not so significant, the best tensile results for this sample may be because of agglomeration but also there are maybe other characteristics that contribute to tensile results?

REPLY: Thanks for the suggestion. Literature reports on the tensile strength were provided in the introduction section (Line 83-95). The improvement in mechanical properties at the interface between mHA1 and PLLA matrix may be attributed to the promotion of stress transfer by the uniformly dispersed high-modulus mHA1, which results in bifurcation and the deflection of microcracks, and consequently facilitates energy dissipation. More importantly, due to the presence of intermolecular stereocomplexation, mHA1 can play a bridging role to prevent the propagation of microcracks at the interface.

COMMENT 18: Figure 7: Please note what on diffractogram means SC, and beta? If you mention reflections in manuscripts 200,110 and 203 please put them in Figure. There are abbreviations in Figure 7 that are not explained in the manuscript please align text with Figure 7.

REPLY: Thanks for the comment. The diffraction peaks for neat PLLA at 2θ=16.8° and 19.0° were attributed to the characteristic peaks of the beta crystal corresponding to the (200/110) and (203) crystal planes, respectively. The diffraction peaks at 2θ = 11.6 °, 20.8°, and 23.9 ° were attributed to sc (stereocomplex) crystallites and corresponded to the (110), (300)/(030), and (220) lattice planes, respectively.

COMMENT 19: In Conclusion, the text should be concluded from the very beginning and put some sentences about good results of obtained samples and then about characterization. The rest of the conclusion should emphasize the novelty of research once again and also put the best results from this study.

REPLY: Thanks for the valuable advice. We have revised this section according to the Reviewer’s suggestion.

Reviewer 2 Report

This is nice contribution from Prof. Chao Zhang group, in this work In this work, his group systematically investigated the Stereocomplexation reinforced high strength poly(L-lactide)/nanohydroxyapatite composites for potential bone repair applications. Thus, the manuscript stands as is, and I recommend publication in Materials after the authors address the following relatively minor comments.

  1. The author should explain and cite some papers in the introduction about related similar materials such as caprolactone/PEG/ Polysaccharides/ APCN mmaterials based reported works. That will enhance the readers understanding and articles visibility well.
  2. I would like to recommend the author should include the control data in the Cell viability experimental data and graph.
  3. Author should add the cytocompatibility Cells images in the manuscript.

Author Response

RESPONSES TO THE REVIEWERS’ COMMENTS

Thank you for your help with our manuscript entitled “Stereocomplexation reinforced high strength poly(L-lactide)/nanohydroxyapatite composites for potential bone repair applications” (manuscript ID: 1570312). We have carefully read the valuable comments, and revised the manuscript accordingly. Please kindly check enclosed revised manuscript, in which corrections/revisions are presented in the Track Change mode. The responses to the reviewers’ comments are listed below for your evaluation.

With regards,

Chao Zhang

Reviewer #2:

COMMENT 1: The author should explain and cite some papers in the introduction about related similar materials such as caprolactone/PEG/ Polysaccharides/ APCN materials based reported works. That will enhance the readers understanding and articles visibility well.

REPLY: Thanks for the suggestion. We have accordingly cited relevant papers and updated the reference list as per the suggestion.

COMMENT 2: I would like to recommend the author should include the control data in the Cell viability experimental data and graph.

REPLY: Thanks for the helpful suggestion. For the cell viability experiment, medical grade HA was regarded as the control and mHA1-3 as the experimental group to verify the safety of mHA1-3. It has been proved that mHA1-3 have appreciable hemocompatibility and cytocompatibility (Figure 11).

COMMENT 3: Author should add the cytocompatibility cells images in the manuscript.

REPLY: Thanks for the comment. We have done the experiment of live/death staining before, and the results show that the material has good cytocompatibility. However, it is difficult to capture high-quality picture in a short time due to the material itself. (The size of PLA/HA films used in cell experiment was 10×10×1mm. After 24 hours of cell culture, the films will deform and it is difficult to focus and take picture in a short time.) The results of Figure.11 can also prove that the material has good cytocompatibility.

Round 2

Reviewer 1 Report

The authors provided full responses for Reviewers and changed the manuscript text according to suggestions.